# Accuracy of BIS monitoring using a novel interface device connecting conventional needle-electrodes and BIS sensors during frontal neurosurgical procedures

Hideki Harada[1]*, Seiya Muta[2], Tatsuyuki Kakuma[3], Misa Ukeda[4], So Ota[4,5], Maiko Hirata[4], Hiroshi Fujioka[6], Osamu Nakashima[2], Barbara Dietel[7], Miyuki Tauchi[7]*

1 Neuroanesthesia Research Laboratory, Cognitive and Molecular Institute of Brain Diseases, Kurume University, Kurume, Japan, 2 Department of Clinical Laboratory Medicine, Kurume University School of Medicine, Kurume, Japan, 3 The Biostatistics Center, Kurume University School of Medicine, Kurume, Japan, 4 Department of Anesthesiology, Kurume University School of Medicine, Kurume, Japan, 5 Omuta City Hospital, Omuta, Japan, 6 Department of Neurosurgery, Kanmon Medical Center, National Hospital Organization (NHO), Yamaguchi, Japan, 7 Department of Medicine 2 –Cardiology and Angiology, Laboratory for Molecular and Experimental Cardiology, Universitätsklinikum Erlangen, Friedrich-Alexander University Erlangen-Nürnberg, Erlangen, Germany

* zxc06612@nifty.com (HH); miyuki.tauchi@uk-erlangen.de (MT)

## Abstract

### Background

Bispectral index (BIS) monitoring is a widely used non-invasive method to monitor the depth of anesthesia. However, in the event of surgeries requiring a frontal approach, placement of the electrode may be impossible at the designated area to achieve a proper BIS measurement.

### Methods

We developed an investigational interface device to connect needle-electrodes to BIS sensors. The safety and clinical performance were investigated in patients who underwent surgery. Direct BIS values from a disposable BIS electrode and indirect values via the interface device were simultaneously recorded from the same areas of electrode placement in a single patient. The agreement between the direct and indirect BIS values was statistically analyzed.

### Results

The interface device with a silver electrode demonstrated sufficient electric conduction to transmit electroencephalogram signals. The overall BIS curves were similar to those of direct BIS monitoring. Direct and indirect BIS values from 18 patients were statistically analyzed using a linear mixed model and a significant concordance was confirmed (indirect BIS = 7.0405 + 0.8286 * direct BIS, p<0.0001). Most observed data (2582/2787 data points, 92.64%) had BIS unit differences of 10 or less.

**Data Availability Statement:** All summarized data are presented in the manuscript. Other data are all kept as raw data with patients' information. Because the size of participants is small and the

study site locates in a small city, there is a privacy concern. Requests can be addressed to the ethics committee, Clinical Research Center, Kurume University Hospital (i_rinri@kurume-u.ac.jp), or to the principal investigator (Hideki Harada).

**Funding:** The authors received no specific funding for this work.

**Competing interests:** The authors have declared that no competing interests exist.

## Conclusions

The interface device provides an opportunity for intraoperative BIS monitoring of patients, whose clinical situation does not permit the placement of conventional adhesive sensors at the standard location.

## Introduction

Bispectral index (BIS) monitoring is widely used to assess the depth of anesthesia [1]. Currently, BIS sensors are placed on the forehead to measure frontal lobe electroencephalogram (EEG) [2]. Surgical stimulations may be detected by EEG responses from frontal areas but not from central, parietal, temporal, or occipital areas [3]. Accordingly, EEG responses, as well as BIS have been shown to be topographically dependent [4]. The most and only reliable area of the sensor placement is the forehead, for which the BIS system has been developed and validated. It is currently considered that the use of other areas for sensor attachment is not easily interchangeable and requires very much caution [5].

In the events of surgeries requiring a frontal approach or concurrent regional oxygen saturation ($rSO_2$) monitoring, optimal placement of the electrode may not be possible to achieve a proper BIS measurement. To enable proper BIS monitoring in such surgeries, we sought to develop an interface device to connect conventional EEG needle-electrodes to BIS sensors.

In the present study, we assessed clinical performance and safety of a novel interface device, which connects the BIS Vista™ system to conventional EEG needle-electrodes, aiming to achieve a reliable BIS monitoring without obstructing frontal surgical procedures or $rSO_2$ monitoring.

## Materials and methods

### Ethical statement

The interface device presented in this manuscript was an investigational device. Written consent was obtained from all patients following detailed information about the study approach. The prospective clinical study was reviewed and approved by the independent ethic committee of the University of Kurume (No. 16050; approved on June 20, 2016; PI: H. Harada) and was carried out in accordance with the Declaration of Helsinki. Following the Japanese guideline for an unapproved medical device in clinical studies [6], patient recruitment was started before registration of the study in a public domain to protect the intellectual property. It was registered at University hospital Medical Information Network (UMIN000031217, February 9. 2018, PI: H. Harada) after the patent was filed. We confirm that all ongoing and related trials for this device are registered. This manuscript adheres to the EQUATOR guidelines (SAMPL and STARD).

### Development of an interface device

**The interface device.** The device was constructed to interface between the subdermal needle-electrodes (NE-220B; Technomed Europe, Maastricht, Netherland) and the Covidien BIS™ Quatro sensor (Minneapolis, MN, USA; Fig 1A–1C). An end of the lead cable (L200) was mounted with a plug connector MS155-S type for the needle-electrodes. Another end was mounted with a plate electrode (diameter 10mm; thickness 0.25mm), made of either stainless steel, silver/ silver chloride (Ag/AgCl), or silver (Ag) to test feasibility. The plate electrodes

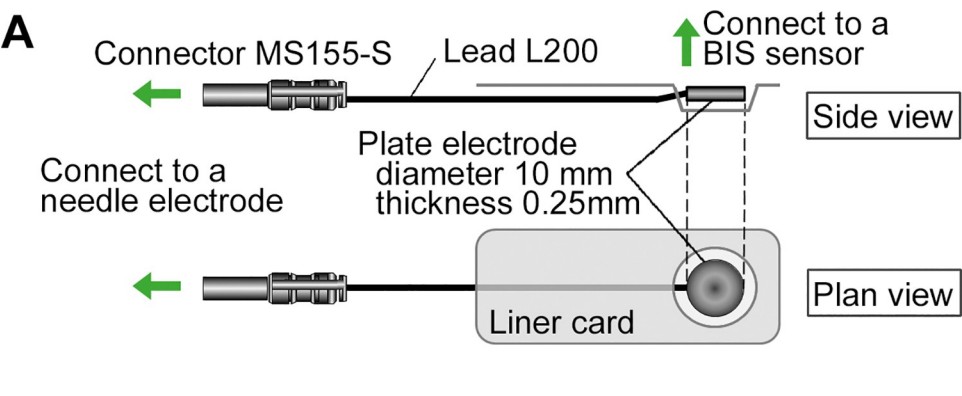

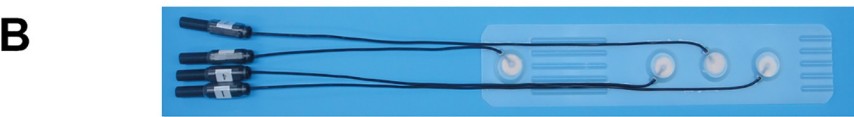

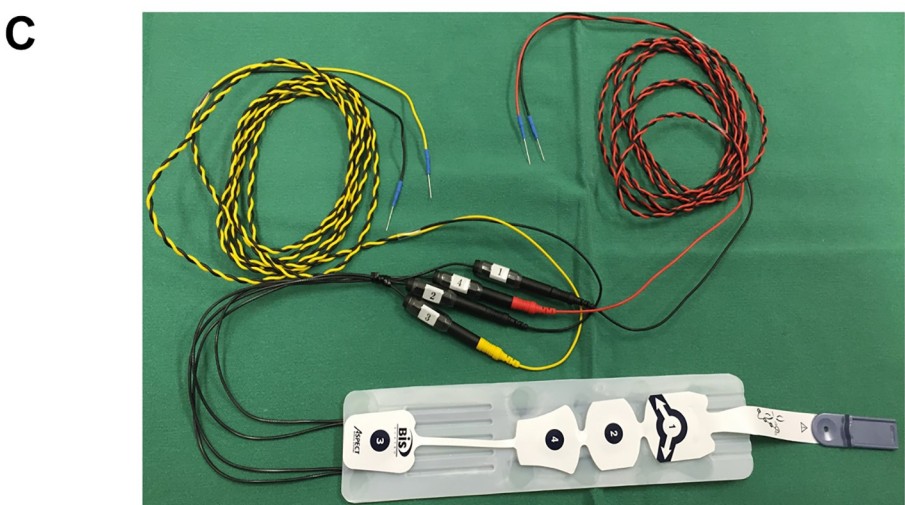

**Fig 1. The interface device.** (A) Specifications of the interface device. (B) The interface device. Four electrodes are integrated into a silicone-coated high-impact polystyrene liner card. (C) The interface device connected to the needle-electrodes and a BIS™ Quatro sensor. BIS: bispectral index.

were integrated into a liner card made of silicone-coated high-impact polystyrene, which lines the disposable BIS™ Quatro sensors in the package. The liner card enables a precise and secure connection of the plate electrodes and a BIS™ Quatro sensor.

**Impedance check.** The BIS monitoring system performs automatic impedance check whether the electrical conduction of sensors is sufficient for EEG analyses. Four electrodes were connected to a BIS™ Quatro sensor via the interface device (Fig 1C), and the needle-electrodes were submerged in saline. The impedance of all needle-electrodes, of the ground electrode, and between two electrodes was measured under a condition of 128Hz/1nA. BIS monitoring system shows "PASS" and BIS can be calculated when following criteria are met: (1) The impedance values of each electrode are less than 7.5 kΩ; (2) the impedance value of the

ground electrode is less than 30 kΩ; and (3) the impedance values between two electrodes (No. 1 vs. 3 and No. 1 vs. 4) are less than 30 kΩ.

**Safety evaluation.** Safety of the device was investigated in compliance with the IEC 60601–1 standard "Medical Electrical Equipment–Part 1: General Requirement for Safety and Essential performance". Leakage tests were performed to preclude any risk for an electrical shock caused by direct contact with the device. Two randomly chosen devices were examined under normal conditions and under single-fault conditions with an interruption of one power supply conductor at a time, with regards to earth leakage current, touch current, patient leakage current (Types BF), and patient auxiliary current. All measurements were done by Nihon Koden Corp. (Kurume, Fukuoka, Japan).

## Clinical studies

**Patients.** Patients were asked for their willingness to participate in the study in a convenience sampling manner when the principal investigator was on duty for their planned neuro- (clinical study 1) or orthopedic (clinical study 2) surgery. Inclusion criteria were ≥20 years old and class I or II of the American Society of Anesthesiologists (ASA) Physical Status Classification. Patients in clinical study 2 with the following conditions were excluded: altered levels of consciousness before the surgery due to intracranial disorders; neurological disorders; psychiatric disorders; pathergy test positive; and metal hypersensitivity. Additionally, patients with injury and/or skin disease on the forehead or surgical procedures impeding BIS sensor attachment at the forehead were excluded, while patients were asked for their agreement to participate in the clinical study 1 only when his/her clinical condition did not allow a conventional BIS monitoring.

**Safety assessment.** All adverse events (AEs) in the entire perioperative period were recorded. Subdermal needle-electrodes used in the study are known occasionally to induce subcutaneous bleeding and infection. These AEs were appropriately treated.

*Clinical study 1*: *BIS measurement using the interface device*. This study was carried out as proof-of-concept without a reference test. Four needle-electrodes were placed on the patient's forehead immediately after anesthesia induction (propofol 2mg/kg, remifentanil 0.25μg/kg per min, and rocuronium 0.6mg/kg). The needle-electrodes were then connected to the BIS monitor via the interface device. The BIS, electromyogram (EMG), and signal quality index (SQI) were recorded throughout the anesthesia maintenance period along with general monitoring of anesthesia.

*Clinical study 2*: *Comparison between BIS values obtained directly from BIS electrode sensor (dBIS) and indirectly from needle-electrodes via the interface device (indBIS)*. The accuracy of the BIS values obtained indirectly via the interface device (indirect recording; indBIS) was assessed by comparing with conventional directly-recorded BIS (dBIS) as the reference standard. Patients received a disposable BIS^TM Bilateral sensor (Covidien) on the forehead following the manufacturer's instruction. The RE and RT electrodes were connected to the interface device. The anesthesia was induced and needle-electrodes were inserted subcutaneously under the adhesive pads of BIS^TM Bilateral LE and LT sensors without direct contact with LE and LT electrodes (Fig 2). The BIS^TM Bilateral sensor was connected to the BIS™ Complete 4-Channel Monitoring System (Covidien) to record signals directly through BIS^TM Bilateral sensors LE and LT and indirectly from the needle-electrodes at the same areas but via the interface device attached to RE and RT. Using this sensor placement strategy, the EEG signals collected from both methods should be nearly identical. After impedance was checked, BIS, EMG, and SQI were recorded throughout the surgery until emergence. Anesthesia was induced with propofol (2mg/kg), remifentanil (0.25μg/kg per min), and rocuronium (0.6mg/kg), and maintained with $O_2$-air-sevoflurane.

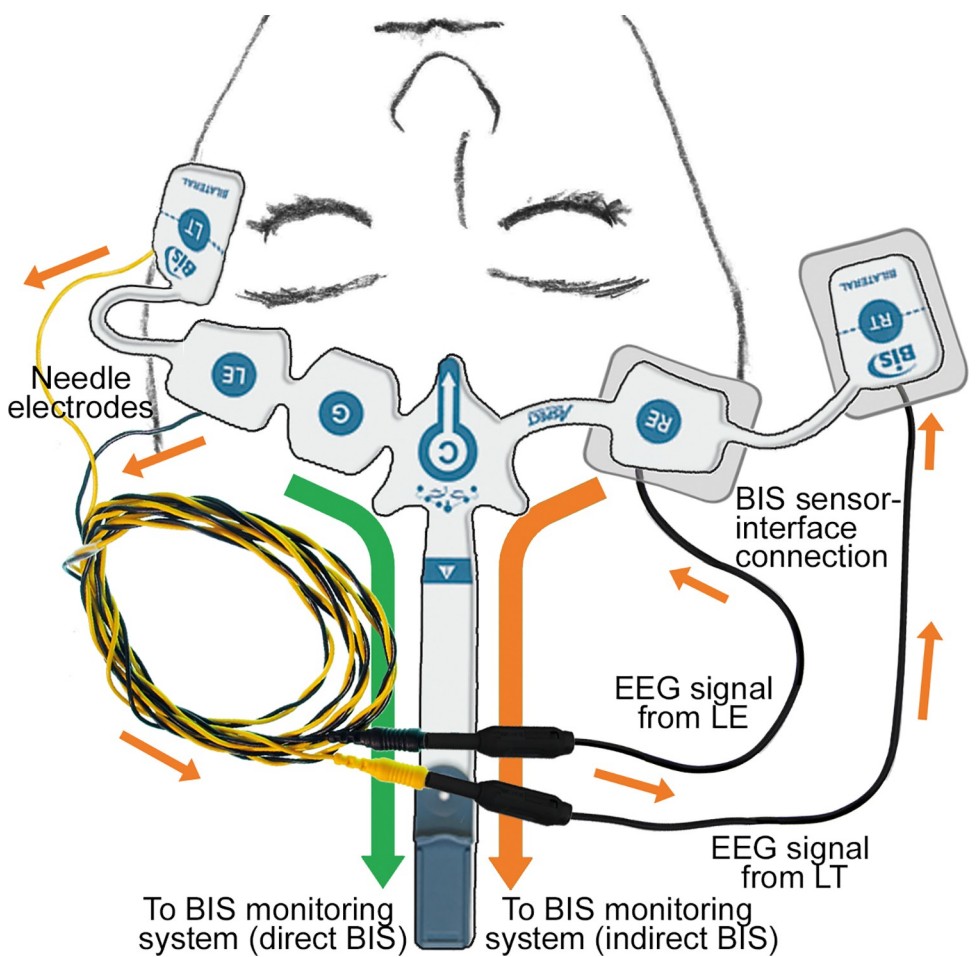

**Fig 2. The interface device setting for direct and indirect recording.** The setting for the clinical study 2 using BIS™ Bilateral sensors. Left BIS™ Bilateral sensor is attached at the forehead following the manufacturer's instruction. Two needle-electrodes are inserted subcutaneously adjacent to the LE and LT electrodes and connected to the RE and RT side via the interface device. Arrows designate the signal paths (orange arrows: indirect bispectral index (BIS) recording; green arrows: direct BIS recording). BIS: bispectral index, EEG: electroencephalogram.

## Statistical analyses

BIS values were automatically stored by BIS VISTA™ software. BIS values of each time point were individually plotted and a correlation coefficient was calculated to evaluate the agreement between dBIS and indBIS. The main object of this study was to examine agreement between two methods. The linear mixed model was used to test concordance between dBIS and indBIS (S1 Text). With regard to sample size and statistical power, formal power analysis was not conducted since there were no clear scientific and clinical guidelines for "equivalence margins" for agreement. Variabilities of measurements among patients were included as random parameters for intercepts and slopes. The adequacy of the model fit was examined using marginal and conditional residuals. Additionally, data were stratified according to the dBIS value (<60 or 60 +) and exploratory analyzed. Tests were two-tailed and a p-value less than 0.05 was considered statistically significant. We set BIS value differences of ±10 between dBIS and indBIS, which were considered clinically acceptable for sake of interpreting results of data analyses.

Bland-Altman analyses were performed to assess the agreement between dBIS and indBIS. In consideration of unequal numbers of BIS observation points in repeated measurements of

continuously changing values, the mixed-effects method was used to estimate the mean bias and the limits of agreement [7] (S1 Text).

All statistical analyses were performed with SAS® software.

## Results

### The interface device with Ag electrode has sufficient electric conduction for BIS measurement

The interface device was constructed with a plate electrode made of three different materials (stainless steel, Ag/AgCl, and Ag). The interface device with a stainless-steel electrode did not pass the check, indicating high impedance that was outside of the measurable range. The Ag/AgCl electrode showed a measurable impedance of around 1000 Ω. However, BIS could not be obtained due to noise, when the interface was connected to the Quatro sensor. The Ag electrode met all the four criteria, and the BIS values were successfully recorded. Only the Ag electrode generated sufficient electric conduction. Therefore, all clinical studies were carried out using the interface device with Ag electrodes.

### IndBIS via the interface device demonstrated a reasonable curve during surgery (clinical study 1)

Six patients underwent the BIS measurement using four needle-electrodes via the interface (Fig 3A, 3B). A representative graph for the values of EMG, SQI, and BIS is shown in Fig 3C. BIS was stable during the surgery [42.5 (2.18); mean (SD), otherwise noted]. SQI was consistently high [96.8 (4.23)] and EMG was consistently low [27.4 (0.51)]. SQI had downward peaks every 10 min, which coincided with the automatic impedance check of the BIS system (Fig 3C; arrows). During the 10-minute recovery period, BIS increased gradually, along with the increase of EMG and the decrease of SQI. The curves of these three parameters were similar in other patients.

### IndBIS via the interface device and dBIS demonstrated a strong agreement (clinical study 2)

A total of 21 patients participated in the study to compare BIS values between direct and indirect measurements. Patients' characteristics are summarized in Table 1.

Data from three patients were excluded from analyses: In two patients, the indBIS was not obtained due to a failed impedance test; and in one patient, BIS was obtained but the impedance of the BIS™ Quatro sensor (LE and LT) was unusually high, affecting BIS values (Fig 4). Signals and corresponding BIS values obtained at the left side (LEFT or LT) were from the BIS™ Bilateral sensor directly. Signals and corresponding BIS values obtained at the right side (RIGHT or RT) were from the same sites on the left forehead, but with the needle-electrodes via the interface device (Fig 2). EEG waves form direct and indirect BIS sensors were reasonably similar (Fig 5A). Representative BIS, EMG, and SQI graphs, superimposing values obtained from direct and indirect sensors in a single patient are shown in Fig 5B–5D, respectively. Both BIS and EMG were very similar and the correlation coefficients were $r^2 = 0.8922$ and 0.9779, respectively, demonstrating a strong agreement (Fig 5B', 5C'). SQI showed regular downward peaks, corresponding to automatic impedance checks by the system (Fig 5D). Although SQI was not very closely matched between two recordings (Fig 5D', $r^2 = 0.6709$), its effect on the BIS correlation was minimal.

Fig 6 shows an exemplary case from a single patient with unstable BIS between 100–200 minutes after recording started (Fig 6A). The unstable BIS was likely an artefact caused by

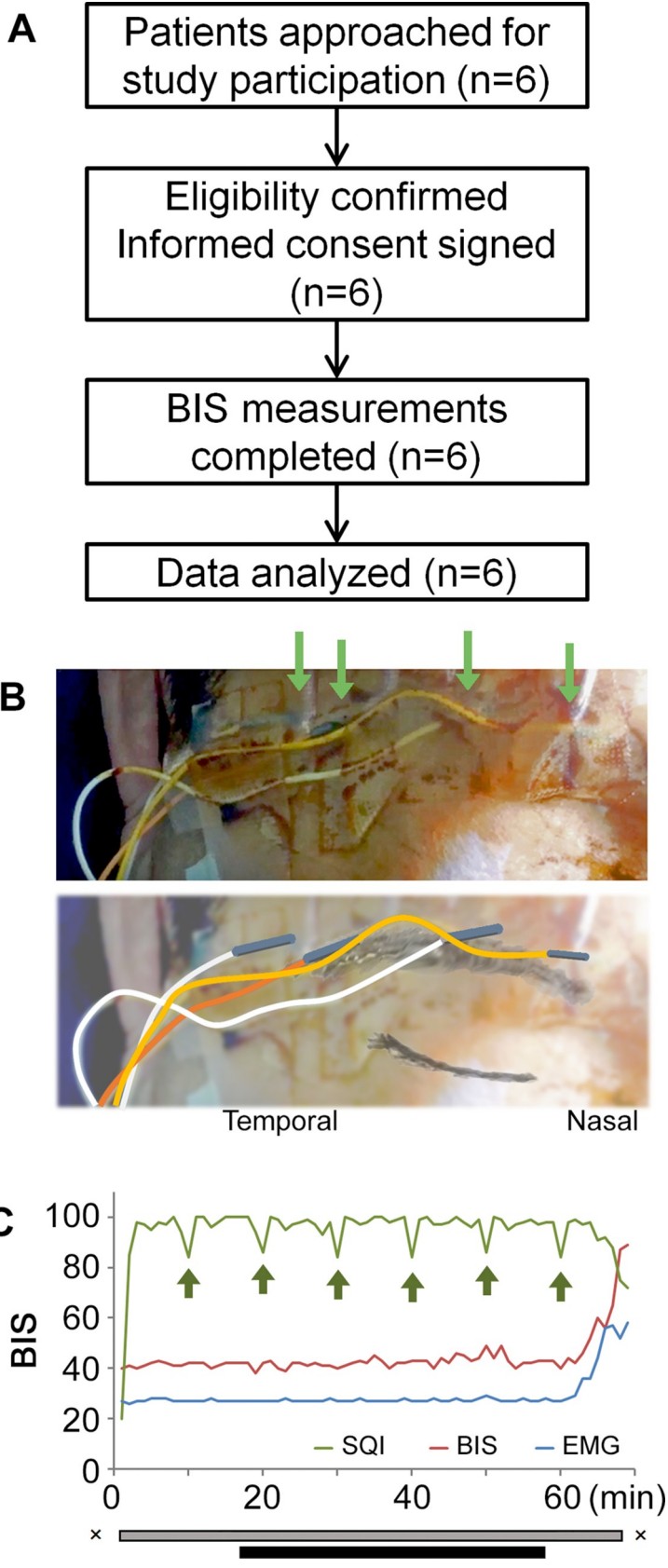

**Fig 3. Indirect BIS recording via the interface device (clinical study 1).** (A) The flow of the participants in clinical study 1. (B) Needle-electrodes placement. Electrodes are inserted in four areas (arrows), where electrodes of a conventional BIS™ Quatro sensor are to be placed. (C) A representative plotting of SQI (green), BIS (red), and EMG (blue). The SQI decreases every 10 minutes, corresponding to the automatic impedance check of the BIS system. The period between X: Anesthesia induction until recovery; grey bar: Intubated period; black bar: Surgical procedure; BIS: bispectral index; EMG: electromyogram; SQI: signal quality index.

unstable EMG signals (Fig 6B) because of intentional stimulations of orbicularis oculi muscle for neuromonitoring purposes. Even with this vigorous EMG disturbance, the dBIS and indBIS in this patient had only a slight discrepancy and were well correlated with the correlation coefficient of $r^2$ = 0.7398 (Fig 6A'). SQI in the direct recording was strongly affected by impedance check and the correlation between SQI in the direct and indirect recording was poor (Fig 6C', $r^2$ = 0.2427). Notably, the poor SQI only slightly influenced the correlation in BIS (Fig 6A').

The linear mixed model included 2786 values (total = 2787) from 18 patients. The indBIS values were predicted as indBIS = 7.0405 + 0.8286 * dBIS (95% CI for the intercept and slope: 3.7410, 10.3399 and 0.7581, 0.8991, respectively) (Fig 7A, red line). Residual analyses showed an adequate model fit and the agreement between dBIS and indBIS values was statistically significant ($p<0.0001$).

In case the stratified data of indBIS <60 from all patients (n = 18) were used, indBIS values were predicted with the formula of indBIS = 21.6025 + 0.493 * dBIS (95% confidence interval (CI), 18.3675 and 24.8445 for the intercept, 0.4057 and 0.5804 for the slope) and the agreement was statistically significant ($p<0.0001$ for both). A total of 5 patients showed significantly deviant estimates of intercept, slope, or both (p-value range between <0.0001 and 0.0328). IndBIS 60+ data from 10 patients were excluded from analysis because they were less than 10% of all data points of the individual. The prediction curve was indBIS = −17.4773 + 1.1776 * dBIS (95% CI, −25.6271 and −9.3276 for the intercept, 1.0883 and 1.2669 for the slope), and indBIS and BIS values were significantly associated (p = 0.0014 for intercept; $p<0.0001$ for slope). The best fitted model was obtained by including both indBIS <60 (n = 18) and 60+ (n = 8) ($p<0.0001$). The prediction curves were indBIS = 13.4156 + 0.6953 * dBIS (95% CI, 9.6541 and 17.1771 for the intercept, 0.6110 and 0.7796 for the slope) for <60, and the slope was 1.2515 for 60+ (Fig 7A, blue line).

The Bland–Altman plot is shown in Fig 7B. Using all data values, the mean bias was estimated as 0.66 (red dotted line) and 95% limits of agreement were 12.13 and −10.80 (red solid lines) (Fig 7B, left panel). The median of the difference between indBIS and dBIS values was 0.00 (interquartile range, 5.00) and the mean was −0.66 (SD, 5.64). Six data points showed a difference of 25 or greater, including the maximum difference of 46. In the 2787 data points

**Table 1. Patients' characteristics[a].**

| Demographic characteristics | | N = 21 |
|---|---|---|
| Sex | Male | 12 patients (57.1%) |
| | Female | 9 patients (42.9%) |
| Age [year] | | 58.70 ± 15.00 (58, 32–79) |
| Height [cm] | | 160.81 ± 8.75 (160.2, 139.7–175.5) |
| Weight [kg] | | 62.56 ± 12.30 (61.2, 38.5–86.6) |
| Duration of monitoring [minutes] [b] | | 153.78 ± 85.06 (126, 61–423) |

[a] Data are expressed as mean ± SD (median, min–max), otherwise noted.

[b] Data from 18 patients whose BIS data were obtained.

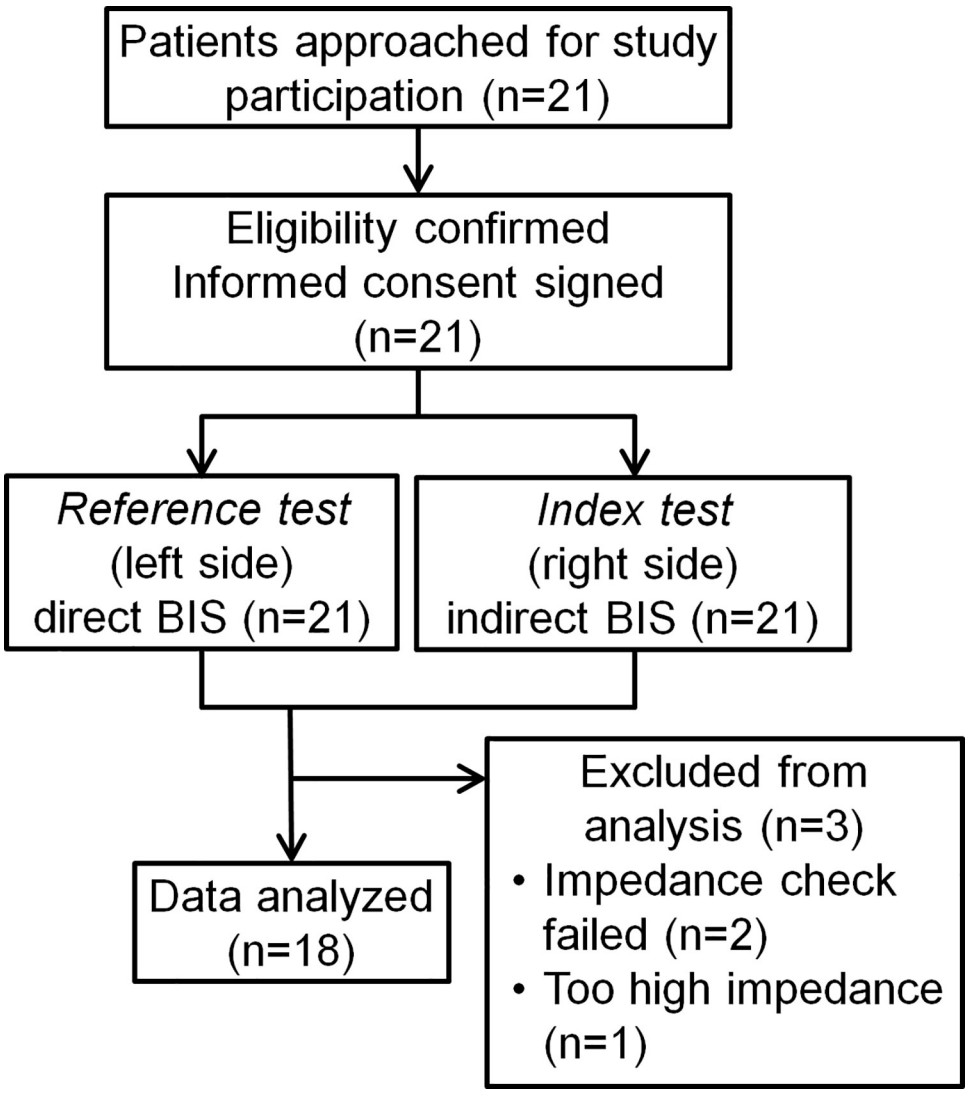

**Fig 4. Flow of the participants in clinical study 2.**

analyzed, 2582 data points (92.64%) demonstrated a difference of less than 10 BIS units between dBIS and indBIS values. In this regard, differences in individual patients seemed to play a role: In 14 patients, 2235/2334 (95.76%) of data points demonstrated the dBIS-indBIS difference of less than 10, while 4 patients showed lower concordance (347/453; 76.60%).

Additionally, the Bland–Altman analyses were carried out in two stratified data (<60 and 60+). The data set of <60 dBIS (n = 2391) had a bias of –0.08 (blue dotted line) and very narrow 95% limits of agreement (–10.04 to 9.88, blue solid lines) (Fig 7B, right panel). Meanwhile, the other data set of 60+ dBIS (n = 395) had broader agreement limits (–9.96 + 20.25) with a higher mean bias (5.14) than those of <60 dBIS (Fig 7B, right panel).

### Safety assessment

No AE was observed during the perioperative period. No patients showed clinical signs of intraoperative awareness. No death occurred from the anesthesia induction until discharge. No major bleeding or subcutaneous bleeding did occur.

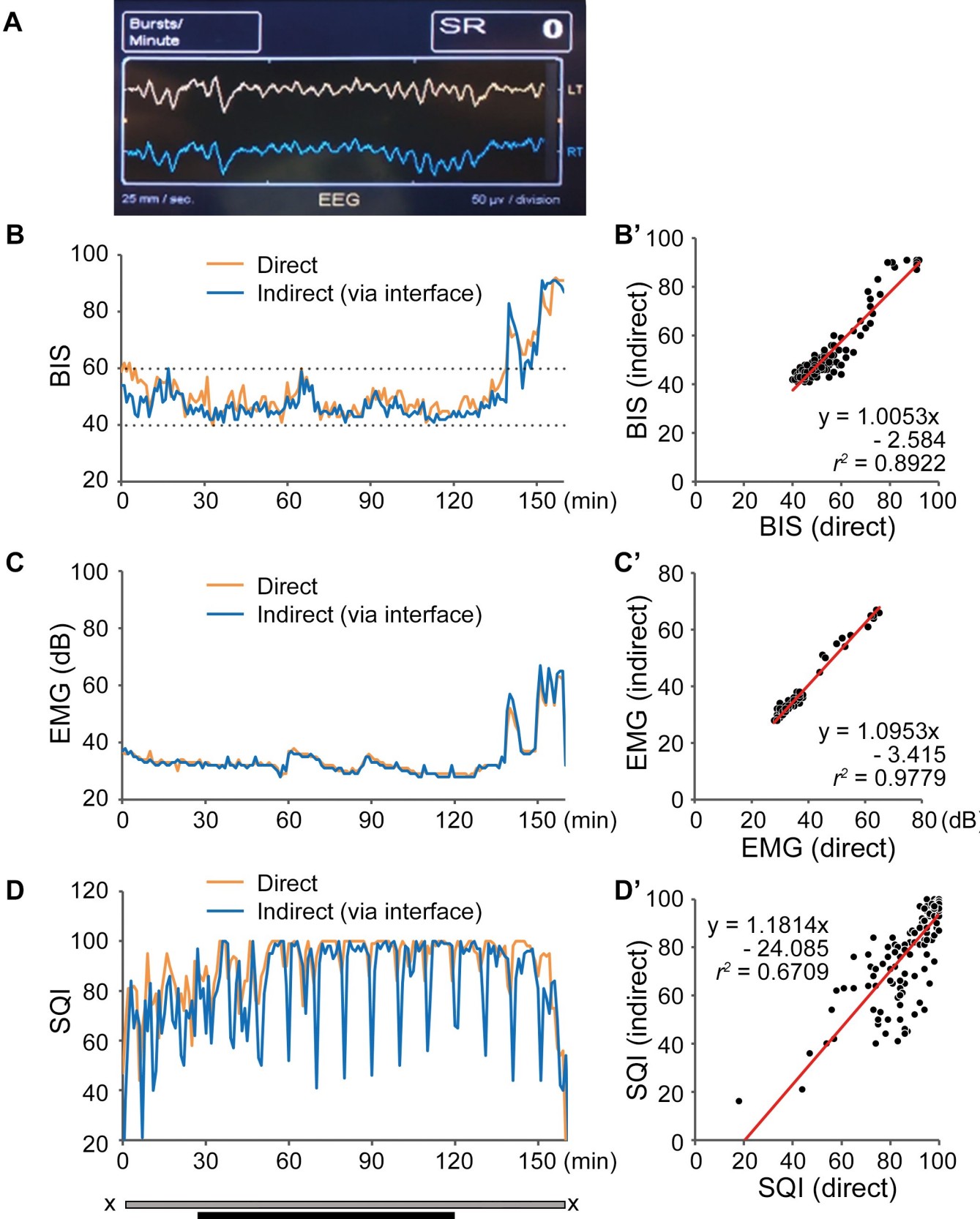

**Fig 5. Results of clinical study 2.** (A) A representative captured image of the BIS monitor. The yellow line represents the EEG wave obtained from the left side (direct recording) and the blue line from the right side (indirect recording via the interface device). These two lines are nearly identical. (B-D) Representative BIS (B), EMG (C), and SQI (D) curves from a single patient with a strong agreement between direct and indirect BIS values. Curves from direct recording (orange lines) and indirect recording (blue lines) are superimposed. B'-D' designate correlation analyses for BIS, EMG, and SQI, respectively. SQI for indirect recording was strongly affected by automatic impedance check every 10 minutes (D, blue line), resulting in the poor correlation coefficient of $r^2$ = 0.6709, but it had a minimal impact on the correlation coefficients in BIS and EMG. The period between X: Anesthesia induction until recovery; grey bar: Intubated period; black bar: Surgical procedure; BIS: bispectral index, EMG: electromyogram, SQI: signal quality index.

## The leakage currents of the interface device were within the tolerance limits

The results of the leakage current measurement are displayed in Table 2. All measured currents were under the tolerance limits regulated by the IEC 60601–1 standard.

## Discussion

We developed and investigated a novel interface device to connect conventional needle-electrodes and BIS monitoring sensors. The device was safe and indBIS values were similar to dBIS values.

This device enables BIS monitoring by collecting EEG signals from the forehead, even with limited space due to clinical situations. BIS monitoring has been investigated to place elsewhere, such as nasal [8], occipital [5, 9], auricular [10, 11], or mandibular [12] areas, and demonstrated a reasonable correlation, with the best concordance by the nasal dorsum measurement [8]. However, BIS has been considered to be topographically dependent [3, 4]. Therefore, validation is largely missing for the use of the forehead sensors at alternative areas and it is currently considered that the use of other areas is not easily interchangeable [5]. Our device largely overcomes the topographical problem. The 95% agreement limit in the present study was about the same or narrower than the nasal placement [8].

The idea to use needle electrodes for BIS measurement has been investigated previously [13, 14]. In these studies, BIS sensors and attached needle electrodes were modified, and tested in human patients or animals to demonstrate its interchangeability to the original sensor. In the present study, the interface device was to connect a commercially available BIS sensor to needle electrodes without modifying them. Using this simple device, the needle electrodes can be easily assembled with the BIS sensors by a practicing anesthetist without the need for special adjustments. Furthermore, the device is for multiple use.

As demonstrated in clinical study 2, dBIS and indBIS values showed a strong agreement. Notably, most of the observed indBIS values (2582/2787, 92.64%) were within the clinically acceptable 10-unit difference [5, 9] from dBIS. The most frequently observed difference between dBIS and indBIS in the entire dataset was 0. The 95% limits of agreement were narrow: Especially in the <60dBIS stratified group, it was –10.04 to 9.88. The difference tended to be bigger in the 60+ data set, which would require attention during clinical use of the interface. Collectively, these data showed that the indBIS is interchangeable to dBIS and clinically tolerable [5, 9].

There were 6 data points out of 2786 with an extremely large difference of 25 or greater (max. 46). All these BIS data were obtained from only two patients with obvious artefacts, such as the patient shown in Fig 6, who received neuromonitoring stimulations. Therefore, in such cases, the discrepancies are easily predictable and even the dBIS values are not to be trusted without other monitoring parameters.

In some patients, we found a large discrepancy in SQI. The reason for that is not clear but we believe from our data that it is not because of the needle electrodes nor the interface device, but because of the proprietary BIS algorism to determine SQI. The raw data from BIS monitoring using BIS^TM Bilateral sensor in VISTA system consistently showed that impedance was

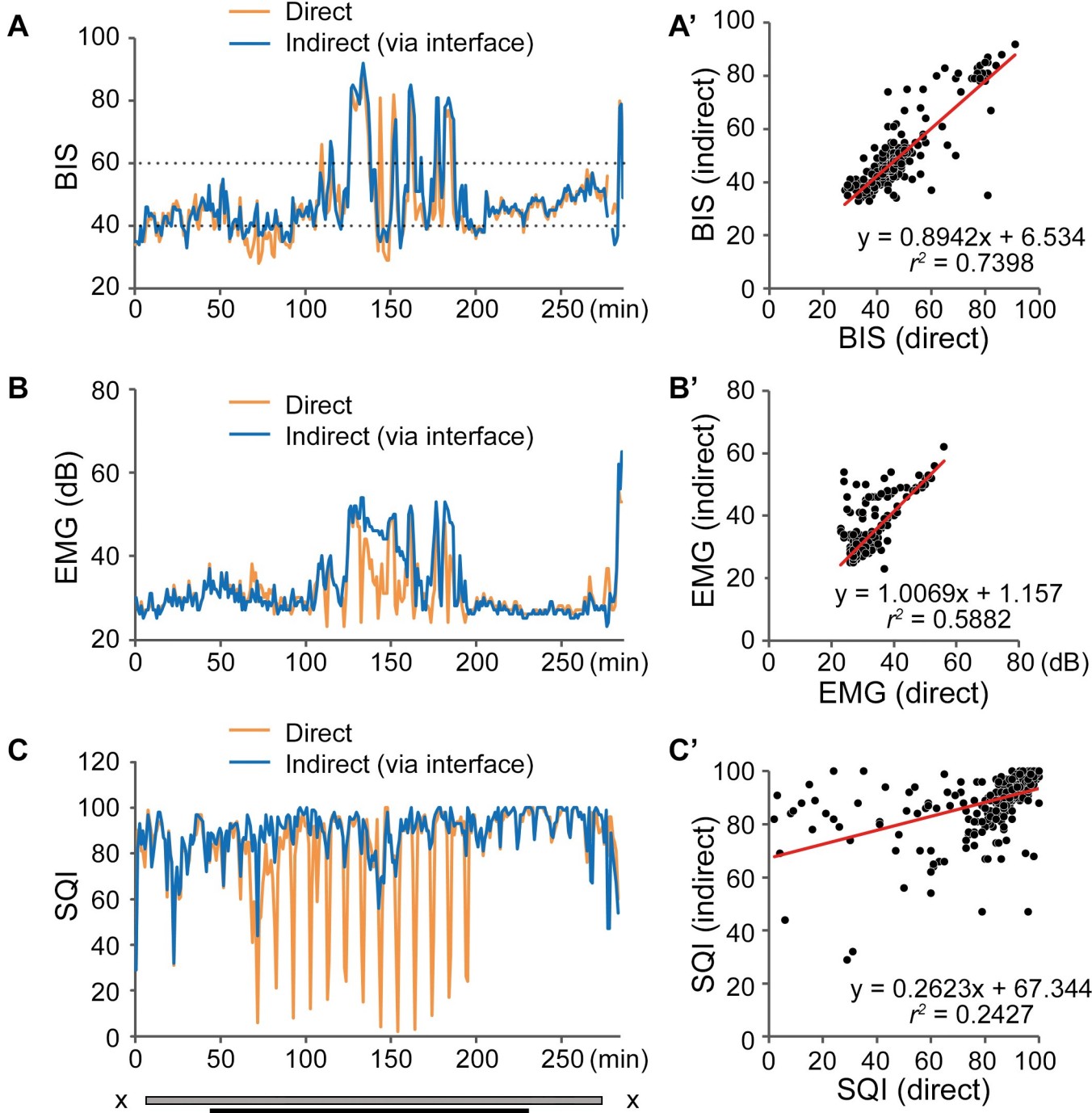

**Fig 6. An example of unstable BIS.** (A-C) Representative BIS (A), EMG (B), and SQI (C) curves from a single patient with an artefact noise in EMG. Curves from direct recording (orange lines) and indirect recording (blue lines) are superimposed. A'-C' panels designate correlation analyses for BIS, EMG, and SQI, respectively. Due to stimulation of orbicularis oculi muscle, EMG was strongly disturbed (B), and the impedance check strongly affected SQI of direct recording during the orbicularis oculi muscle stimulation (C). Nevertheless, the effect on the BIS agreement was minimal (A'). The period between X: Anesthesia induction until recovery; grey bar: Intubated period; black bar: Surgical procedure period; BIS: bispectral index, EMG: electromyogram, SQI: signal quality index.

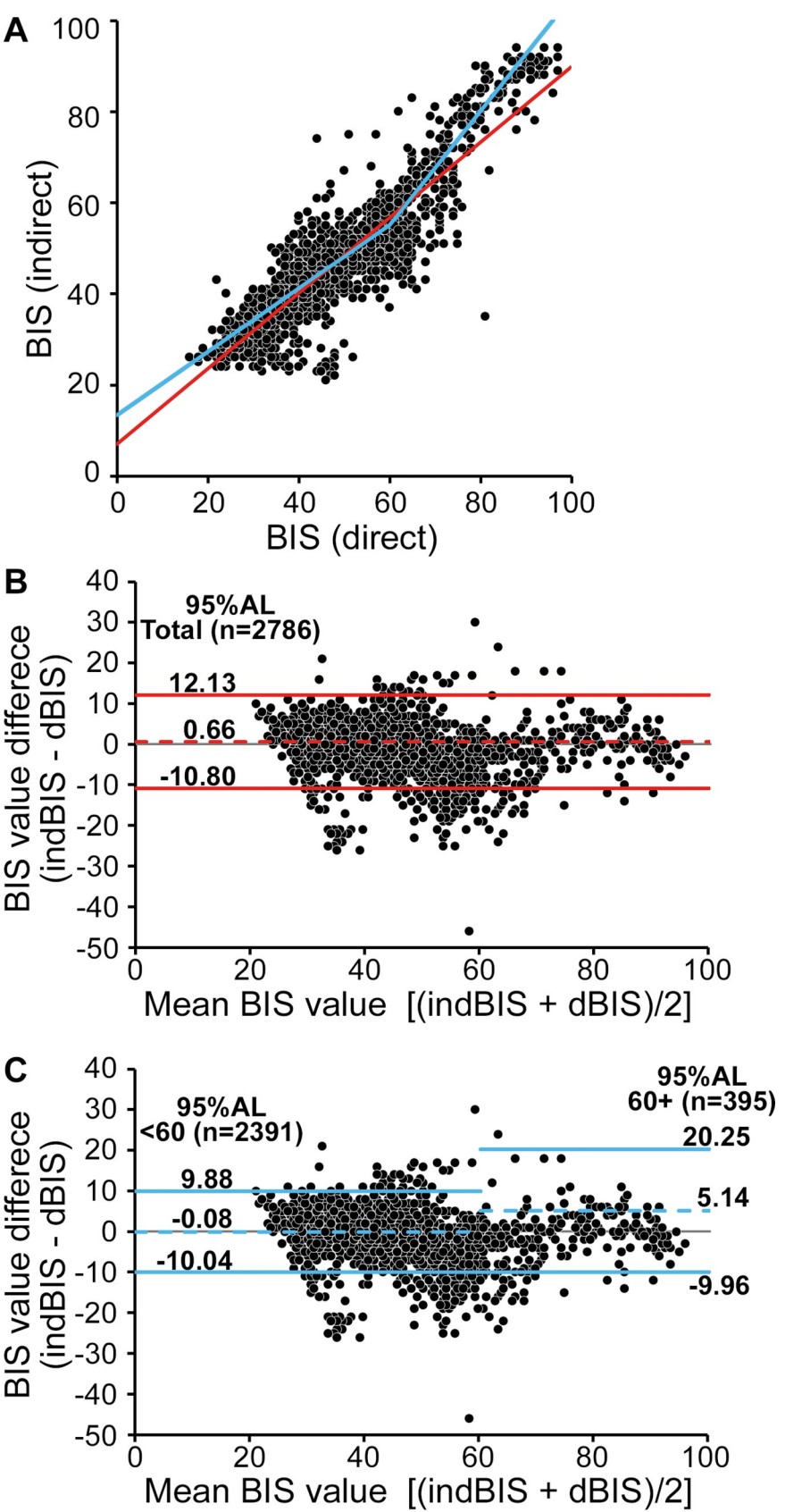

**Fig 7. Association between dBIS and indBIS.** (A) Plotting of all BIS values analyzed (2786 pairs of observations, including 2391 with BIS<60 and 395 with BIS ≥60). Red line: the prediction equation determined by the linear mixed model analysis for all data: indBIS = 7.0405 + 0.8286 * dBIS. Blue line: The prediction equation determined by the linear mixed model using stratified data: indBIS = 13.4156 + 0.6953 * dBIS (<60) and indBIS = −19.9564 +1.2515 * dBIS (60+). (B) and (C) Bland-Altman plot. The bias and 95% agreement limits (95%AL) estimated using all data are designated in red (B). The 95% ALs are narrower if it is estimated using dBIS < 60 than those of 60+ (blue lines, C). The dotted lines represent bias and solid lines represent 95% ALs. BIS: bispectral index, ALs: agreement limits.

over the highest limit exactly every 10 minutes for 1 minute at three out of four electrodes. It is obviously an artificial pattern: therefore, we consider that the extremely high impedance is not an actual value, but an artificial one because of the automatic impedance checks. This timing coincides with the SQI spikes, indicating that SQI values are apparently dependent on the impedance values.

The impedance check lowers SQI more prominently in indBIS than in dBIS, probably because of the SQI calculation algorism. As written above, the impedance values exceeding the upper measurement limit are observed only in three out of four electrodes, regardless of the type of electrodes. The SQI for indBIS was based on two electrodes with unusually high impedance every 10 minutes but SQI for dBIS was based on one unaffected electrode and one electrode with unusually high impedance. Based on the facts described above, we found two supporting data sets: First, the SQI spikes were small in SQI data recorded using needle electrodes and the interface device only, being connected to a BIS™ Quatro sensor (Fig 3C). Next, we examined SQI data, recorded by using the BIS™ Bilateral sensor (no needle electrodes, no interface device), from several patients. We confirmed that, in all cases, the impedance values exceeding the upper measurement limit were observed only in three out of four electrodes. When SQI are plotted for left- and right sides separately, there was a SQI discrepancy between the two SQIs, in most cases with larger SQI spikes at right side recording, which was used for indBIS recording.

From above observation, we concluded that the observed SQI spikes are not needle- or interface device-specific. In either case, the SQI spikes were not substantially influencing the BIS values as a fact from the data. With the one-side recording using BIS™ Quatro sensor, the effect is minimal, which is equivalent for the real-world practice.

**Table 2. Leakage currents.**

| Current | | Device 1 (μA) | Device 2 (μA) | Tolerance limit (μA) |
|---|---|---|---|---|
| Earth leakage current | Normal condition | 47 | 49 | ≤ 5000 |
| | Single-fault condition | 74 | 79 | ≤ 10000 |
| Touch current | Normal condition | 6 | 6 | ≤ 100 |
| | Single-fault condition | 1 | 1 | ≤ 500 |
| Patient leakage current (Type BF applied part) | Normal condition, DC | 0 | 0 | ≤ 10 |
| | Normal condition, AC | 1 | 1 | ≤ 100 |
| | Single-fault condition, DC | 0 | 0 | ≤ 50 |
| | Single-fault condition, AC | 1 | 1 | ≤ 500 |
| Patient auxiliary current | Normal condition, DC | 0 | 0 | ≤ 10 |
| | Normal condition, AC | 0 | 0 | ≤ 10 |
| | Single-fault condition, DC | 0 | 0 | ≤ 50 |
| | Single-fault condition, AC | 0 | 0 | ≤ 50 |

BF: body floating; DC: direct current; AC: alternating current.

Increased or unstable EMG activities influence BIS values [15]. In some patients (including the patient shown in Fig 6), EMG became unstable to some extent. This affected BIS values from both indirect and direct recordings. The disagreement between dBIS and indBIS was prominent in EMG ($r^2$ = 0.5882), but moderate in BIS values. Indeed, BIS values showed a reasonably high correlation coefficient ($r^2$ = 0.7398). The possible reasons for unstable EMG were mechanical artefacts, such as the movement of lead cables, surgical processes, impedance check, and patients' movement. Although the discrepancy demands careful use of the interface device and great attention during surgery, EMG affects BIS obtained from the conventional adhesive sensor as well. Therefore, the unstable EMG-driven unstable BIS values are not necessarily an interface device-specific problem.

The observed small disagreement between dBIS and indBIS may not be caused by the interface device but the tolerance margins existing in BIS algorithms, and thus maybe unavoidable. In a previous study, BIS was monitored from a single patient using two BIS™ Quatro sensors and two recording machines [16]. They observed a discrepancy in two recordings and concluded that the failed reproducibility was because of the BIS system itself. We made effort to minimize potential bias seen in the study (i.e., two recording machines and adjacent but different areas of sensor placement): We placed the adhesive electrodes and the needle-electrodes directly next to each other in every patient and simultaneously recorded using a single BIS VISTA™ system to enable a precise comparison of two measures. The discrepancy we observed was smaller than that in the previous study [16], which let us consider that the disagreement between dBIS and indBIS were within the intrinsic tolerance margins and clinically not significant.

There is room for improvement in the interface device. The key improvement would be to obtain a constantly good SQI, which could be done by improving the adhesion of interface electrodes to the BIS conventional sensors. We used recycled liner cards, which were delivered with conventional BIS sensors and detached once from it. The two excluded cases from clinical study 2 were very likely due to poor adhesion between the interface device and the BIS™ Quatro sensor, resulting in high impedance. Slight differences in the attachment may account for the between-patients' differences in the signal quality we observed.

The interface device can be connected not only to needle-electrodes but also commercially available plate electrodes for EEG measurements. The plate electrodes cannot be fixed so secure as the needle-electrodes, but require a small space for placement as well, and are non-invasive. Therefore, in case the electrodes are accessible by an anesthesiologist to re-assure the adhesion, the plate electrodes may be advantageous.

In conclusion, this investigational interface device provides an opportunity for intraoperative BIS monitoring of patients, whose clinical situation does not permit the use of conventional adhesive BIS sensors. There was a small discrepancy between BIS values in direct and indirect measurements, but it can be overcome with careful monitoring of SQI and EMG, also in conjunction with other available clinical signs for the intraoperative monitoring of anesthesia. The clinical benefit of the interface device is of great significance in monitoring anesthesia.

## Supporting information

**S1 Checklist.**
(PDF)

**S2 Checklist.**
(PDF)

**S1 Text. Statistical models.**
(DOCX)

**S1 File.**
(PDF)

**S2 File.**
(PDF)

## Acknowledgments

We thank all patients who agreed to participate in the study.

## Author Contributions

**Conceptualization:** Hideki Harada.

**Data curation:** Seiya Muta, Misa Ukeda, Maiko Hirata, Osamu Nakashima.

**Formal analysis:** Hideki Harada, Seiya Muta, Tatsuyuki Kakuma, Miyuki Tauchi.

**Investigation:** Hideki Harada, Seiya Muta, So Ota, Maiko Hirata, Osamu Nakashima.

**Methodology:** Hideki Harada.

**Project administration:** Hideki Harada.

**Resources:** Hideki Harada.

**Supervision:** Hideki Harada.

**Validation:** Hiroshi Fujioka.

**Visualization:** Miyuki Tauchi.

**Writing – original draft:** Hideki Harada, Tatsuyuki Kakuma, Miyuki Tauchi.

**Writing – review & editing:** Tatsuyuki Kakuma, Barbara Dietel, Miyuki Tauchi.

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
