## [Decision Letter · Decision Letter 0]

29 Apr 2021

PONE-D-21-02149

Accuracy of BIS monitoring using a novel interface device connecting conventional needle-electrodes and BIS sensors during frontal neurosurgical procedures

PLOS ONE

Dear Dr. Harada,

Thank you for submitting your manuscript to PLOS ONE. After careful consideration, we feel that it has merit but does not fully meet PLOS ONE’s publication criteria as it currently stands. Therefore, we invite you to submit a revised version of the manuscript that addresses the points raised during the review process.

Although interesting, the manuscript presents great limitations. In particular statistical analysis need to be improved. Please carefully address all Reviewers' comments, in particular the comments of Reviewer #4. 

We look forward to receiving your revised manuscript.

Kind regards,

Laura Pasin

Academic Editor

PLOS ONE

Journal Requirements:

2. Thank you for submitting your clinical trial to PLOS ONE and for providing the name of the registry and the registration number. The information in the registry entry suggests that your trial was registered after patient recruitment began. PLOS ONE strongly encourages authors to register all trials before recruiting the first participant in a study.

1) your reasons for your delay in registering this study (after enrolment of participants started);

2) confirmation that all related trials are registered by stating: “The authors confirm that all ongoing and related trials for this drug/intervention are registered”.

3. Please include captions for *all* your Supporting Information files at the end of your manuscript, and update any in-text citations to match accordingly. Please see our Supporting Information guidelines for more information: http://journals.plos.org/plosone/s/supporting-information.

4. We note that Figure [S2.BIS setting with the interface device (video)] includes an image of a patient in the study. 

Reviewers' comments:

Reviewer's Responses to Questions

**Comments to the Author**

1. Is the manuscript technically sound, and do the data support the conclusions?

Reviewer #1: Yes

Reviewer #2: Yes

Reviewer #3: Yes

Reviewer #4: No

2. Has the statistical analysis been performed appropriately and rigorously? 

Reviewer #1: Yes

Reviewer #2: Yes

Reviewer #3: Yes

Reviewer #4: No

3. Have the authors made all data underlying the findings in their manuscript fully available?

Reviewer #1: Yes

Reviewer #2: Yes

Reviewer #3: No

Reviewer #4: Yes

4. Is the manuscript presented in an intelligible fashion and written in standard English?

Reviewer #1: Yes

Reviewer #2: Yes

Reviewer #3: Yes

Reviewer #4: Yes

5. Review Comments to the Author

Reviewer #1: Dear author, I congratulate for your work.

I have no comment. Your paper is appropriate and I believe room there is minimum space for further improvement.

I could only suggest to provide a better image 3B that appear to be blurry (too much light?)

Reviewer #2: In this manuscript, the authors demonstrated a novel interface device that connects conventional needle-electrodes to BIS monitoring sensors, thereby enabling the indirect BIS monitoring without topographical problems. This, otherwise, would be impossible to achieve using any of existing BIS sensors whose validation is largely relies on the use of it on the forehead. This interface device showed a good agreement between direct and indirect BIS values (2582/2787; 92.64%) within a clinically acceptable 10-unit difference. The concept and the experimental demonstration are interesting which I believe this manuscript is acceptable for publication in this journal if the followings can be addressed.

1. In this study, the authors insist that they developed the novel interface device, which connects the BIS Vista system to conventional EEG needle-electrodes for indirect BIS monitoring. I wonder that this interface device is novel enough. This interface system is constructed by connecting two components that are commercially available (BIS sensor and needle electrodes). In addition, the studies are already reported showing the use of needle electrode for BIS monitoring (T. M. Hemmerling et al., Anesth Analg, 2002, 95:1675-7, S. A. Greene et al., Comp. Med. 2002, 52:424-8). Thus, it is recommended for the authors to provide the novelty of this study compared to the previously reported works.

2. In Figure 5D’ and Figure 6C, even though SQI value was not very closely matched between indirect and direct recordings, the authors claim that their effect on the BIS correlation was minimal. Why is there little correlation? Besides, SQI value not for direct recording but for indirect recording was only strongly affected by automatic impedance check (Figure 5D). Why is the indirect recording only vulnerable to automatic impedance check? Are those recording results reliable with low value of SQI?

3. By the way, there are typing errors:

- In page 21, line 393-394, EGM should be corrected to EMG.

- In Figure 7B, Y axis, “differece” should be corrected to “difference”.

Reviewer #3: The paper was well presented. The data supported the conclusions. A minor revision is needed. The authors did not make the data fully accessible. Only the results of statistical analyses are available in the manuscript.

Reviewer #4: The statistics are mainly regression with linear mixed models and the Bland Altman procedure for method comparison. These appear to be applied appropriately. However, there are major concerns.

1. The study lacks a reasonable statistical hypothesis for which the sample size of 18 to 21 should be justified in terms of the alpha level and anticipated power, even if the unit of analysis is otherwise. The entire design is not clear. Why so many points on Figure 7? This statistical justification should be provided.

2. The Bland Altman procedure (Figure 7) is suspect and poorly explained. There is a great bulk of points beyond what appears to be the lower (+10, -10) bias limit. This limit is not very meaningful as presented with so many points violating those limits especially on the low side. This is not convincing for agreement. Also, what exactly are the 95% limits and how do they actually affect the overall conclusion?

3. Figure 6, A prime does not support the conclusion of the paper. There does not appear to be an acceptable level of agreement of the indirect and direct BIS. What is the hypothesis test of the slope and intercept on this line, especially the test for intercept equal to 0 and the slope equal to one?

6. PLOS authors have the option to publish the peer review history of their article (what does this mean?). If published, this will include your full peer review and any attached files.

Reviewer #1: **Yes: **Alessandro De Cassai

Reviewer #2: **Yes: **Chi Hwan Lee

Reviewer #3: No

Reviewer #4: No

---

## [Author Response · Author response to Decision Letter 0]

1 Jun 2021

Dear Dr. Pasin,

Thank you very much for the review comments. We were very pleased to receive a chance to resubmit our manuscript and appreciate you and all reviewers for helping us to improve the manuscript. All comments were taken seriously, the manuscript was revised thoroughly, and we resubmit herewith our revised manuscript. Please find a separate letter, which include our point-by-point responses to the reviewers.

We would thank you very much for your consideration to publish our revised manuscript in PLOS ONE and hope the revised manuscript reached your criteria to accept it for publication.

Sincerely,

Miyuki Tauchi and Hideki Harada, on behalf of all coauthors

---

## [Decision Letter · Decision Letter 1]

21 Jul 2021

PONE-D-21-02149R1

Accuracy of BIS monitoring using a novel interface device connecting conventional needle-electrodes and BIS sensors during frontal neurosurgical procedures

PLOS ONE

Dear Dr. Harada,

Thank you for submitting your manuscript to PLOS ONE. After careful consideration, we have decided that your manuscript does not meet our criteria for publication and must therefore be rejected.

I apologize for the long time you had to wait. Unfortunately it was very, very difficult to find reviewers.

My decision is based on the lack of formal testing on the estimates of intercept and slope, as pointed out by Reviewer #4. 

I am sorry that we cannot be more positive on this occasion, but hope that you appreciate the reasons for this decision.

Yours sincerely,

Laura Pasin

Academic Editor

PLOS ONE

Reviewers' comments:

Reviewer's Responses to Questions

**Comments to the Author**

1. If the authors have adequately addressed your comments raised in a previous round of review and you feel that this manuscript is now acceptable for publication, you may indicate that here to bypass the “Comments to the Author” section, enter your conflict of interest statement in the “Confidential to Editor” section, and submit your "Accept" recommendation.

Reviewer #4: (No Response)

2. Is the manuscript technically sound, and do the data support the conclusions?

Reviewer #4: No

3. Has the statistical analysis been performed appropriately and rigorously? 

Reviewer #4: No

4. Have the authors made all data underlying the findings in their manuscript fully available?

Reviewer #4: Yes

5. Is the manuscript presented in an intelligible fashion and written in standard English?

Reviewer #4: Yes

6. Review Comments to the Author

Reviewer #4: With respect to the BIS regressions, the reviewer asked:

What is the hypothesis test of the slope and intercept on this line, especially the test for intercept equal to 0 and the slope equal to one?

The authors responded:

No formal testing was performed on the estimates of intercept and slope. Of note, these data, although we believe the unstable BIS is an artefact because of neuromonitoring, are included in the formal testing described in the manuscript and plotted in Fig 7.

Such may be the case. However, an R-square less than 0.9 when considering agreement is not that convincing. At best, the agreement is weak, statistically.

In Figure 7A you want the intercept to be 0 and the slope to be one. A formal test not rejecting these null hypotheses that intercept=0 and slope=1 would convince one of agreement. That was the reason for requesting a formal test.

7. PLOS authors have the option to publish the peer review history of their article (what does this mean?). If published, this will include your full peer review and any attached files.

Reviewer #4: No

- - - - -

---

## [Author Response · Author response to Decision Letter 1]

19 Aug 2021

Dear Dr. Pasin,

Dear editors,

We are re-submitting our manuscript. We received a disappointing decision letter for our revised manuscript in July. We went through the review comments to understand the problem in the manuscript, but we could not convince ourselves from the review comments that our manuscript did not meet the quality required by PLOS ONE. We appreciate that you accepted our appeal and reconsider the manuscript after.

Herewith we submit our point-by-point response (as a separate file), and the manuscrips with and without tracking changes. We do not have any new changes in the manuscript since the last submission: the tracked changes are from the version after the first review.

Sincerely,

Hideki Harada

Miyuki Tauchi

---

## [Editor Report · Decision Letter 2]

4 Oct 2021

Accuracy of BIS monitoring using a novel interface device connecting conventional needle-electrodes and BIS sensors during frontal neurosurgical procedures

PONE-D-21-02149R2

Dear Dr. Harada,

We’re pleased to inform you that your manuscript has been judged scientifically suitable for publication and will be formally accepted for publication once it meets all outstanding technical requirements.

Kind regards,

Laura Pasin

Academic Editor

PLOS ONE

Additional Editor Comments (optional):

Thank you for your email and additional response to reviewers. I acknowledge I did not fully understand the previous revision. I'm really sorry. Now everything is clear and I'm happy to accept your manuscript. 
---

## [Editor Report · Acceptance letter]

11 Oct 2021

PONE-D-21-02149R2 

Accuracy of BIS monitoring using a novel interface device connecting conventional needle-electrodes and BIS sensors during frontal neurosurgical procedures 

Dear Dr. Harada:

I'm pleased to inform you that your manuscript has been deemed suitable for publication in PLOS ONE. Congratulations! Your manuscript is now with our production department. 

Kind regards, 

on behalf of

Dr. Laura Pasin 

Academic Editor

PLOS ONE